# Two Novel Species of *Talaromyces* Discovered in a Karst Cave in the Satun UNESCO Global Geopark of Southern Thailand

**DOI:** 10.3390/jof8080825

**Published:** 2022-08-07

**Authors:** Salilaporn Nuankaew, Charuwan Chuaseeharonnachai, Sita Preedanon, Sayanh Somrithipol, Supicha Saengkaewsuk, Papichaya Kwantong, Sarinya Phookongchai, Prasert Srikitikulchai, Noppol Kobmoo, Xin-Cun Wang, Zhi-Feng Zhang, Lei Cai, Satinee Suetrong, Nattawut Boonyuen

**Affiliations:** 1Plant Microbe Interaction Research Team (APMT), National Center for Genetic Engineering and Biotechnology (BIOTEC), National Science and Technology Development Agency (NSTDA), Khlong Nueng, Khlong Luang, Pathum Thani 12120, Thailand; 2National Biobank of Thailand (NBT), National Science and Technology Development Agency (NSTDA), Khlong Nueng, Khlong Luang, Pathum Thani 12120, Thailand; 3State Key Laboratory of Mycology, Institute of Microbiology, Chinese Academy of Sciences (CAS), Beijing 100101, China; 4Southern Marine Science and Engineering Guangdong Laboratory, Guangzhou 511458, China

**Keywords:** section *Trachyspermi*, polyphasic taxonomy, *Trichocomaceae*, cave-dwelling soil micro-fungi, Phu Pha Phet karst cave

## Abstract

Karst caves are oligotrophic environments that appear to support a high diversity of fungi. Studies of fungi in Thailand’s caves are limited. During a 2019 exploration of the mycobiota associated with soil samples from a karst cave, namely, Phu Pha Phet in the Satun UNESCO Global Geopark in Satun Province, southern Thailand, two previously undescribed fungi belonging to *Talaromyces* (*Trichocomaceae*, *Eurotiales*, *Eurotiomycetes*) were studied using a polyphasic approach combining phenotypic and molecular data. Based on datasets of four loci (ITS, *BenA*, *CaM*, and *RPB2*), phylogenetic trees of the section *Trachyspermi* were constructed, and two new species—*Talaromyces phuphaphetensis* sp. nov. and *T. satunensis* sp. nov.—phylogenetically related to *T. subericola, T. resinae,* and *T. brasiliensis,* are described. Detailed descriptions and illustrations of the new species are provided. This study increases the number of cave-dwelling soil fungi discovered in Thailand’s Satun UNESCO Global Geopark, which appears to be a unique environment with a high potential for discovering fungal species previously undescribed.

## 1. Introduction

The genus *Talaromyces* was introduced [1] with *Talaromyces vermiculatus* (=*T*. *flavus*) as the type of species. *Talaromyces* taxa are classified into *Aspergillaceae*, *Eurotiales*, *Eurotiomycetidae*, *Eurotiomycetes*, Pezizomycotina, and Ascomycota (MycoBank. 2022; Species Fungorum. 2022; accessed on 1 June 2022). This genus is well-known and among the most prevalent groups of fungi, found in a range of habitats, including soil, vegetation, air, living or decaying plants, indoor environments, and a wide range of food products [2,3,4,5,6,7]. Phu Pha Phet Cave, a part of a mycological diversity project associated with Satun Geopark, Thailand’s first UNESCO Global Geopark, is also known as “Diamond Mountain Cave”. It is the fourth largest cavern on earth and the largest cave in Thailand, covering more than 80,000 m^2^. Based on estimated visitation, the cave has been opened as a tourist attraction and is regarded as an anthropogenic disturbance; nonetheless, some areas in the Phu Pha Phet Cave remain closed [8]. Research on fungal diversity and mycological systematics in karst caves has been scarce in Thailand’s Satun UNESCO Global Geopark.

In this study, soil samples randomly obtained from the Phu Pha Phet Cave were subjected to phenotypic examination and phylogenetic approaches, and two new cave-dwelling soil micro-fungi belonging to *Talaromyces*—*T. phuphaphetensis* and *T. satunensis* spp. nov.—were described and compared with similar taxa.

## 2. Materials and Methods

### 2.1. Collection, Isolation, and Morphology

On 3 December 2019, collections were performed during a fungal survey of Phu Pha Phet Cave. Two strains of *Talaromyces* were isolated from soil samples (110 m elevation; 7°07′35″ N 99°59′49″ E) in Thungwa, Manang District, La-Ngu, Satun Province, southern Thailand. Ten or twenty grams of soil were randomly collected at shallow depths (1–5 cm) after removing the surface layer, placed in zip lock bags, preserved at 4 °C in an ice box during collection, and transferred to the mycological laboratory at the National Center for Genetic Engineering and Biotechnology (BIOTEC).

The dilution plate technique was carried out using a modified version of the method of Zhang et al. [9], and 1 g of the sample was suspended in 9 mL of sterile distilled water and then serially diluted 10-fold. Dilutions from 10^−1^ to 10^−5^ were prepared, and 100 µL of each dilution was spread on potato dextrose agar (PDA; Difco, GA, USA) containing two antibiotics (50 μg/mL of ampicillin and 50 μg/mL of streptomycin) with three replicates. Plate cultures were incubated at room temperature for two–three days to allow fungal growth before subculture onto PDA without antibiotics for additional morphological investigation.

After seven days, macroscopic features and growth rates were examined on seven traditional culture media (Czapek yeast autolysate agar (CYA), Czapek’s agar (CZ), malt extract agar (MEA), yeast extract sucrose agar (YES), dichloran 18% glycerol agar (DG18), creatine sucrose agar (CREA), and oatmeal agar (OA, Difco)), as previously described [10]. Strains were inoculated with spore suspensions at three points and incubated in the dark at 25 °C, with additional temperatures of 30 and 37 °C for CYA. Extended incubation of MEA and OA plates for four weeks was performed to observe sexual reproduction. Microscopic observations were carried out on 7-day-old MEA, CZ, and CYA media. Ethanol (70%) and lactic acid (60%) were used to wash excess of conidia and mount slides, respectively. 

Microscopic characters (i.e., conidiophores, conidiogenous cells, and conidia) were examined with a light microscope (OLYMPUS CX31; Olympus Corporation, Japan) and photographed using a Nomarski differential interference contrast microscope (OLYMPUS DP70). The Methuen Handbook of Color created color codes that were used to categorize the observed colors of the colonies [11]. The types and strains were deposited into the Thailand Bioresource Research Center (TBRC; https://www.tbrcnetwork.org, accessed on 21 July 2022) under the names *Talaromyces phuphaphetensis* sp. nov. (TBRC 16281) and *T*. *satunensis* sp. nov. (TBRC 16246). The type specimens are kept in the FUNGARIUM BIOTEC Bangkok Herbarium (BBH; https://www.nbt-microbe.org, accessed on 15 June 2022) as *T*. *phuphaphetensis* BBH 49306 (holotype) and *T*. *satunensis* BBH 49305 (holotype). The MycoBank numbers were registered as *T*. *phuphaphetensis* MB 844613 and *T*. *satunensis* MB 844614.

### 2.2. DNA Extraction, PCR Amplification, and Phylogenetic Analyses

Following the protocols of Sri-indrasutdhi et al. [12], genomic DNA was extracted from 7-day-old cultures grown on MEA using the cetyltrimethylammonium bromide (CTAB) method. The internal transcribed spacer (ITS) region, β-tubulin (*BenA*), calmodulin (*CaM*), and RNA polymerase II (*RPB2*) genes were amplified. The primers and amplification profiles used are shown in Table 1. PCR products were purified and sequenced by Macrogen Inc. (Seoul, South Korea) using the same PCR primers used for PCR amplification. The obtained sequences of ITS, *BenA*, *CaM*, and *RPB2* were assembled and trimmed at both ends in BioEdit v.7.1.3 [13]. The newly generated sequences were deposited in GenBank (the National Centre for Biotechnology Information (NCBI)), and representative *Talaromyces* in the section *Trachyspermi* used in phylogenetic analyses, and their accession numbers are provided in Table 2.

Multiple sequence alignments were performed separately using MAFFT v.7.490 [14] for each locus and adjusted manually. The four datasets were concatenated in BioEdit v.7.1.3 [13]. Maximum likelihood (ML) phylogenetic analyses, including 1000 bootstrap replicates, were performed using RAxML-NG [15] under the GTR + GAMMA model with default parameters on the Debian Linux operating system. Bayesian inference (BI) was carried out using MrBayes v.3.2.7 [16] with 5,000,000 Markov chain Monte Carlo (MCMC) generations, with the first 2,000,000 discarded as burn-in. The consensus tree was visualized and adjusted in Adobe Photoshop 2021 using FigTree v1.4.4 (http://tree.bio.ed.ac.uk/software/figtree, accessed on 10 September 2019).

**Table 1 jof-08-00825-t001:** Molecular markers, primers, and amplification profiles used and generated in this study.

Molecular Locus	Primer Name	Direction	Reference	Amplification Profile		
				Denature	Repeat Step	Extension
Internal transcribed spacers (ITS)	ITS1	Forward	[17]	94 °C (5 min)	35 cycles, 94 °C (45 s), 55 °C (45 s), 72 °C (60 s)	72 °C (7 min)
	ITS5					
	ITS4	Reverse				
β-tubulin (*BenA*)	Bt2a	Forward	[18]	94 °C (10 min)	35 cycles, 94 °C (30 s), 57 °C (30 s), 72 °C (30 s)	72 °C (10 min)
	Bt2b	Reverse				
Calmodulin (*CaM*)	cmd5	Forward	[19]	94 °C (3 min)	30 cycles, 94 °C (1 min), 57 °C (1 min), 72 °C (1 min)	72 °C (10 min)
	cmd6	Reverse				
RNA polymerase II (*RPB2*)	5F2	Forward	[20]	94 °C (3 min)	34 cycles, 94 °C (1 min), 54 °C (1 min), 72 °C (1.30 min)	72 °C (8 min)
	7cR	Reverse				

**Table 2 jof-08-00825-t002:** *Talaromyces* species of sect. *Trachyspermi* used in phylogenetic analyses and their GenBank accession numbers.

Taxon	Original Strain Number	GenBank Accession Number			
		ITS	*BenA*	*CaM*	*RPB2*
*Talaromyces aerius*	CBS 140611 ^T^	KU866647	KU866835	KU866731	KU866991
*T. affinitatimellis*	CBS 143840 ^T^	LT906543	LT906552	LT906549	LT906546
*T. africanus*	CBS 147340 ^T^ = DTO 179-C5	OK339610	OK338782	OK338808	OK338833
*T. albisclerotius*	CBS 141839 ^T^ = DTO 340-G5	MN864276	MN863345	MN863322	MN863334
*T. albobiverticillius*	CBS 133440 ^T^	HQ605705	KF114778	KJ885258	KM023310
*T. amyrossmaniae*	NFCCI 1919 ^T^	MH909062	MH909064	MH909068	MH909066
*T. assiutensis*	CBS 147.78 ^T^	JN899323	KJ865720	KJ885260	KM023305
*T. atroroseus*	CBS 133442 ^T^	KF114747	KF114789	KJ775418	KM023288
*T. austrocalifornicus*	CBS 644.95 ^T^	JN899357	KJ865732	KJ885261	MN969147
*T. basipetosporus*	CBS 143836 ^T^ = FMR 9720	LT906542	LT906563	-	LT906545
*T. brasiliensis*	URM 7618 ^T^	MF278323	LT855560	LT855563	MN969198
*T. calidominioluteus*	CBS 147313 ^T^ = DTO 052-G3	OK339612	OK338786	OK338817	OK338837
*T. catalonicus*	CBS 143039 ^T^ = FMR 16441	LT899793	LT898318	LT899775	LT899811
*T. chongqingensis*	CS26-67 ^T^	MZ358001	MZ361343	MZ361350	MZ361357
*T. clemensii*	PPRI 26753 ^T^	MK951940	MK951833	MK951906	MN418451
*T. convolutus*	CBS 100537 ^T^	JN899330	KF114773	MN969316	JN121414
*T. diversus*	CBS 320.48 ^T^	KJ865740	KJ865723	KJ885268	KM023285
*T. erythromellis*	CBS 644.80 ^T^	JN899383	HQ156945	KJ885270	KM023290
*T. gaditanus*	CBS 169.81 ^T^ = DTO 228-B8	MH861318	OK338775	OK338802	OK338827
*T. germanicus*	CBS 147314 ^T^ = DTO 055-D1	OK339619	OK338799	OK338812	OK338845
*T. guatemalensis*	CCF 6215 ^T^	MN322789	MN329687	MN329688	MN329689
*T. halophytorum*	KACC 48127 ^T^	MH725786	MH729367	MK111426	MK111427
*T. heiheensis*	HMAS 248789 ^T^ = CGMCC 3.18012	KX447526	KX447525	KX447532	KX447529
*T. minioluteus*	CBS 642.68 ^T^	JN899346	MN969409	KJ885273	JF417443
*T. minnesotensis*	CBS 142381 ^T^	LT558966	LT559083	LT795604	LT795605
*T. pernambucoensis*	URM 6894 ^T^	LR535947	LR535945	LR535946	LR535948
** *T. phuphaphetensis* **	**TBRC 16281 ^T^**	**ON692803**	**ON706960**	**ON706962**	**ON706964**
*T. resinae*	CBS 324.83 ^T^ = IMI 080450	MT079858	MN969442	MT066184	MN969221
*T. rubrifaciens*	CGMCC 3.17658 ^T^	KR855658	KR855648	KR855653	KR855663
*T. samsonii*	CBS 137.84 ^T^ = DTO 304-C3 = DTO 169-G6	MH861709	OK338798	OK338824	OK338844
** *T. satunensis* **	**TBRC 16246 ^T^**	**ON692804**	**ON706961**	**ON706963**	**-**
*T. solicola*	DAOM 241015 ^T^	FJ160264	GU385731	KJ885279	KM023295
*T. speluncarum*	CBS 143844 ^T^ = FMR 16671	LT985890	LT985901	LT985906	LT985911
*T. subericola*	CBS 144322 ^T^ = FMR 15656	LT985888	LT985899	LT985904	LT985909
*T. systylus*	BAFCcult 3419 ^T^	KP026917	KR233838	KR233837	-
*T. trachyspermus*	CBS 373.48 ^T^ = IMI 040043	JN899354	KF114803	KJ885281	JF417432
*T. ucrainicus*	CBS 162.67 ^T^ = FRR 3462	JN899394	KF114771	KJ885282	KM023289
*T. udagawae*	CBS 579.72 ^T^ = IMI 197482	JN899350	KF114796	KX961260	MN969148
*T. flavus*	CBS 310.38 ^T^	JN899360	JX494302	KF741949	JF417426

New taxa proposed in this study are in bold. ^T^, Ex-type strain. -, Data not available. Acronyms of culture collections: BAFC/BAFCcult, Culture Collection of the Department of Biological Sciences, Faculty of Exact and Natural Sciences, University of Buenos Aires, Argentina; BCC, BIOTEC Culture Collection, Pathum Thani, Thailand; CBS, Centraalbureau voor Schimmelcultures, CBS-KNAW Culture, Utrecht, Netherlands; CCF, Culture Collection of Fungi, Department of Botany, Faculty of Science, Charles University, Prague, Czech Republic; CGMCC, China General Microbiological Culture Collection Center, Beijing, China; DAOM, Canadian Collection of Fungal Cultures, Ottawa, Canada; DTO, culture collection of Food and Indoor Mycology Group of Westerdijk Institute, Utrecht, Netherlands; FMR, Faculty of Medicine in Reus, Spain; HMAS, Herbarium Mycologicum Academiae Sinicae, Beijing, China; IMI, International Mycological Institute (CABI Bioscience, Eggham), UK; KACC, Korean Agricultural Culture Collection, South Korea; NFCCI, National Fungal Culture Collection of India, India; PPRI, ARC-Plant Protection Research Institute, National Collection of Fungi: Culture Collection, Denmark; TBRC, Thailand Bioresource Research Center, Pathum Thani, Thailand; URM, Universidade Federal de Pernambuco Herbário, Brazil.

## 3. Results

### 3.1. Phylogenetic Analysis

The phylogenetic trees of ITS, *BenA*, *CaM*, and *RPB2* constructed separately using ML analyses and the concatenated datasets of four loci based on ML and Bayesian analyses revealed the relationships among the novel strains (TBRC 16281 and TBRC 16246) and *Talaromyces* species of the section *Trachyspermi* (Figure 1, Figure 2 and Figure 3). Based on the single-gene analyses, our two proposed new species, *T. phuphaphetensis* and *T. satunensis*, were clustered with *T. brasiliensis* URM 7618, *T. resinae* CBS 324.83, and *T. subericola* CBS 144322. The two new species and *T. subericola* formed a monophyletic group, and were revealed as phylogenetically related to *T. brasiliensis* and *T. resinae* (Figure 1 and Figure 2). 

In the ITS and *CaM* phylograms, *T. subericola* was a sister taxon to *T. phuphaphetensis*, and these two lineages were closely related to *T. satunensis* with a good bootstrap support (Figure 1 and Figure 2). In the *BenA* phylogram, our two new species clustered together with a low support value (bootstrap value < 70%) and were closely related to *T. subericola* on a highly supported branch (99%). In the *RPB2* analyses (no sequence data of *T. satunensis*), *T. subericola* was the closest sister taxon to *T. phuphaphetensis*, with good bootstrap support (90%).

Based on the combined datasets of ITS, *BenA*, *CaM*, and *RPB2*, the phylogenetic relationships showed a topology similar to those obtained from each gene individually (Figure 3). The two new species, *T. phuphaphetensis* and *T. satunensis*, formed two single branches and a well-supported clade with *T. brasiliensis*, *T. resinae*, and *T. subericola* (BS/PP = 95%/1.00). *Talaromyces subericola* was a sister taxon of *T. phuphaphetensis*, and these two species were closely related to *T. satunensis* in both fully supported subclades (BS/PP = 100%/1.00). Phylogenetically, *T. resinae* and *T. brasiliensis* were at a basal position, located on a single branch within the same clade as *T. phuphaphetensis* and *T. satunensis*.

### 3.2. Taxonomy

*Talaromyces phuphaphetensis* Nuankaew, Chuaseehar. & Somrith., sp. nov. is shown in Figure 4.

MycoBank: 844613.

Etymology: The specific epithet refers to “Phu Pha Phet Cave”, where the type strain was first collected.

Typification: Thailand, Satun Province, Manang District, Satun UNESCO Global Geopark, Phu Pha Phet cave, from soil, 3 December 2019, Nattawut Boonyuen, Prasert Srikitikulchai and Sita Preedanon, culture, Sita Preedanon, CV00299 (holotype BBH 49306, ex-type strain TBRC 16281).

GenBank numbers: *BenA* = ON706960, *CaM* = ON706962, ITS = ON692803, RPB2 = ON706964.

In: *Talaromyces* sect. *Trachyspermi.*

Colony diameter (7 days, in mm): CYA 8–9; CYA 30 °C 3–5; CYA 37 °C 3–4; CZ 3–4; MEA 16–18; OA 10–12; DG18 8–9; YES 6–7; CREA 3–4.

Colony characteristics: CYA at 25 °C after 7 days: Colonies slightly raised at centers; margins low, entire (<1 mm); mycelia white; texture floccose; sparse to absent sporulation after 21 days; conidia en masse grayish green (25C4); soluble pigment light yellow (2A4); exudates absent; reverse center grayish yellow (2C4) and yellowish white (4A2). MEA at 25 °C after 7 days: Colonies slightly raised at centers; margins low, entire (<1 mm); mycelia white; texture loosely funiculose and floccose; sporulation strong; conidia en masse grayish green (27E4); soluble pigment absent; exudates absent; reverse center orange-gray (5B2) fading into orange-white (5A2). CZ at 25 °C after 7 days: Colonies low, slightly raised at centers; margin entire (2–3 mm); mycelia white; texture velvety; sporulation moderately; conidia en masse dull green (28D4); soluble pigment yellow (2A6) after 14 days of incubation; exudates absent; reverse yellowish gray (3B2). DG18 at 25 °C after 7 days: Colonies low, plane; margins low, plane, entire (2–3 mm); mycelia white; texture velvety; sporulation absent; soluble pigment yellow (2A6) after 14 days of incubation; exudates absent; reverse center orange-white (5A2) fading into pale orange (5A3). OA at 25 °C after 7 days: Colonies slightly raised at centers; margins low, plane, entire (3–4 mm); mycelia white; texture loosely funiculose; sporulation moderate; conidia en masse grayish green (26D3); soluble pigment absent; exudates absent. YES at 25 °C after 7 days: Colonies slightly raised at center, slightly concave, wrinkled; margins low, entire (<1 mm); mycelia white; texture floccose; sporulation absent; soluble pigment absent; exudates absent; reverse grayish yellow (2C4). CREA at 25 °C after 7 days: Acid production absent; poorly growing.

Micromorphology: On MEA, conidiophores mostly biverticillate, minor proportion monoverticillate; stipes finely tuberculate, non-vesiculate, 15–60 × 2.5–3 μm; metulae (2–) 3–6 per stipe, adpressed, 5–9 × 1.5–3 μm; phialides 3–5 per metula, acerose, 7–9.5 × 2–3 μm; conidia globose to sub-globose, smooth-walled, 2–3.5 μm in diameter. Ascomata absent.

Note: Phylogenetically, *Talaromyces phuphaphetensis* falls into a terminal clade of section *Trachyspermi*, including species *T*. *brasiliensis*, *T*. *resinae*, *T*. *satunensis* (described here), and *T*. *subericola* (Figure 3). *Talaromyces phuphaphetensis* and *T*. *satunensis* can be differentiated as having tuberculate stipes and smooth-walled conidia, whereas the other three related species have smooth-walled stipes and conidial ornamentation [4,7,21]. *Talaromyces phuphaphetensis* mainly differs from *T*. *satunensis* in having shorter stipes (15–60 × 2.5–3 μm in *T*. *phuphaphetensis* vs. 20–290 × 2–3.2 μm in *T*. *satunensis*), producing yellow diffusible pigments on CYA after 7 days, CZ and DG18 after 14 days, and possessing strong sporulation on MEA.

In addition, *T*. *phuphaphetensis* showed poor growth on CYA incubated at 37 °C (3–4 mm, 7 days), while *T. brasiliensis*, *T. satunensis,* and *T*. *subericola* had no growth on the medium. Morphological comparisons of *T*. *phuphaphetensis*, *T*. *satunensis,* and the three related species are shown in Table 3.

*Talaromyces satunensis* Nuankaew, Chuaseehar. & Somrith., sp. nov. is shown in Figure 5.

MycoBank: 844614.

Etymology: The specific epithet refers to “Satun”, the name of the province where the species originated.

Typification: Thailand, Satun Province, Manang District, Satun UNESCO Global Geopark, Phu Pha Phet cave, from soil, 3 December 2019, Nattawut Boonyuen, Prasert Srikitikulchai and Sita Preedanon, culture, Sita Preedanon, CV00055 (holotype BBH 49305, ex-type strain TBRC 16246).

GenBank numbers: *BenA* = ON706961, *CaM* = ON706963, ITS = ON692804.

In: *Talaromyces* sect. *Trachyspermi.*

Colony diameter (7 days, in mm): CYA 5–6; CYA 30 °C 4–5; CYA 37 °C No growth; CZ 4–5; MEA 18–20; OA 8–10; DG18 12–13; YES 4–5; CREA 3–4.

Colony characteristics: CYA at 25 °C after 7 days: Colonies raised, sulcate; margins low, entire (<1 mm); mycelia pale gray to yellowish gray; texture floccose; sporulation poorly after 21 days; soluble pigment absent; exudates absent; reverse grayish yellow (4C3). MEA at 25 °C after 7 days: Colonies slightly raised at centers; margins low, plane, entire (1–2 mm); mycelia white; texture floccose; sporulation poor; conidia en masse grayish green (30C3); soluble pigment absent; exudates absent; reverse grayish yellow (3B3) with center fading into orange-white to pale orange (5A2–5A3). CZ at 25 °C after 7 days: Colonies low, plane; margin entire (<1 mm); mycelia white; texture velvety; sporulation moderately; conidia en masse dull green (30D5); soluble pigment absent; exudates absent; reverse yellowish gray (4B2). DG18 at 25 °C after 7 days: Colonies low, plane; margins low, plane, entire (2–3 mm); mycelia white; texture velvety; sporulation absent; soluble pigment absent; exudates absent; reverse yellowish white (2A2). OA at 25 °C after 7 days: Colonies slightly raised at centers; margins low, plane, entire (<1 mm); mycelia white; texture loosely funiculose; sporulation moderate; conidia en masse grayish gray (26D3); soluble pigment absent; exudates absent. YES at 25 °C after 7 days: colonies slightly raised at center, slightly concave; margins narrow (<1 mm); mycelia white; texture velvety; sporulation absent; soluble pigment absent; exudates absent; reverse pale orange (5A3) and grayish orange (5B3). CREA at 25 °C after 7 days: Acid production absent; poorly growing. 

Micromorphology: On MEA, conidiophores mostly biverticillate, minor proportion monoverticillate and terverticillate; stipes tuberculate, non-vesiculate, 20–290 × 2–3.2 μm; metulae 2–5 per stipe, rather adpressed, 5.5–10 × 2–3.3 μm; phialides (2–)3–5 per metula, ampulliform to acerose, 6–9 × 2–3.2 μm; conidia globose to sub-globose, smooth-walled, 2.5–3 μm in diameter. Ascomata absent.

Note: Phylogenetically, *T*. *satunensis* is located within a terminal clade, and it is closely related to *T*. *phuphaphetensis* and *T*. *subericola* (Figure 3). *Talaromyces subericola* differs from our two new species in producing smooth-walled stipes and verruculose conidia. In comparison, *T*. *satunensis* differs from *T. phuphaphetensis* in the absence of diffusible pigments on CYA, lacking growth on CYA at 37 °C, and poor sporulation on MEA. In addition, *T*. *satunensis* has longer stipes and sometimes produces terverticillate branches (see Table 3).

## 4. Discussion

In this study, phylogenies and morphological characters supported the establishment of *Talaromyces phuphaphetensis* and *T*. *satunensis* as two new species belonging to *Talaromyces* section *Trachyspermi*. Phylogenetic analyses based on single loci (i.e., ITS, *BenA*, *CaM*, and *RPB2*) and the multi-locus approach showed that *T*. *phuphaphetensis* and *T. satunensis* are members of the *Talaromyces* clade composed of *T. brasiliensis*, *T. resinae*, and *T*. *subericola*. All phylogenetic trees also indicated that *T*. *subericola* has the closest relationship with our two new species described herein. Based on the combined dataset, the phylogenetic analyses revealed that *T*. *brasiliensis* and *T*. *resinae* are basal to *T*. *phuphaphetensis*, *T. satunensis*, and *T*. *subericola* (Figure 3). 

The topology of the *CaM* tree for *T*. *brasiliensis* and *T*. *resinae* showed a slightly different position (Figure 2). In addition, the species relationships within the section *Tra*-chyspermi, as shown in the phylogenetic trees inferred from *CaM,* were different from those in the trees based on the ITS, *BenA*, and *RPB2* genes. These data are congruent with the studies of Rajeshkumar et al. [5] and Zhang et al. [22]. However, the phylogenetic tree of *CaM* gene sequences could distinguish *T*. *phuphaphetensis* and *T. satunensis* from other species in the section. Although the *RPB2* gene is formally accepted as a potential molecular locus for identifying *Talaromyces* species, it is often difficult to amplify the targeted DNA region [23,24,25]. Unfortunately, this study did not obtain *RPB2* sequence data from *T*. *satunensis*, although we attempted with different PCR profiles. Nonetheless, the phylogenies based on single genes and the concatenated data also confirmed the taxonomic placements of *T*. *phuphaphetensis* and *T. satunensis* as two distinct species in the *Trachyspermi* section.

Both *T. phuphaphetensis* and *T*. *satunensis* are characterized by the production of biverticillate conidiophores, tuberculate-walled stipes, and smooth-walled conidia. They grow restrictedly on CYA, YES, and DG18, slightly faster on MEA, and poorly on CREA. These data are in agreement with the description of the section [25,26]. Colonies of *T*. *phuphaphetensis* produce yellow pigment on CYA, CZ, and DG18. Generally, *Talaromyces* species are reported to be good pigment producers [27,28,29]. Many species in the section *Trachyspermi* (such as *T. albobiverticillius*, *T*. *atroroseus*, and *T*. *minioluteus*) can produce a large amount of red pigment. However, only *T*. *atroroseus* produces pigments without known mycotoxins, which might be suitable for application in the food or healthcare industry as an alternative synthetic dye [27]. Likewise, the new species we propose can serve as an alternative source of natural pigments that need to be investigated for mycotoxin production, enhanced pigment production, and other testing for future research.

## 5. Conclusions

Two isolates of soil fungi were discovered in the Phu Pha Phet Cave of the Satun UNESCO Global Geopark in southern Thailand and identified as part of the genus *Talaromyces* in the section *Trachyspermi*. The two isolates are proposed as new species, namely *Talaromyces phuphaphetensis* and *T*. *satunensis*, based on their morphological and phylogenetic differences from the other species described in the section *Trachyspermi*. The discovery will support future evaluations of the unique species’ potential applications and functions. Information on the mycological biodiversity and habitat of UNESCO’s Satun cave would promote awareness of sustainable conservation and exploitation, supporting the future planning, monitoring, and management of Thai caves in achieving a balance between conservation and development. Furthermore, the results contribute to the knowledge of cave-dwelling soil fungi, their ecological uniqueness and diversity in Thailand, and their global geographical distribution. Interestingly, it is also possible that more new species will be discovered in this peculiar environment in Thailand’s Satun UNESCO Global Geopark.

## Figures and Tables

**Figure 1 jof-08-00825-f001:**
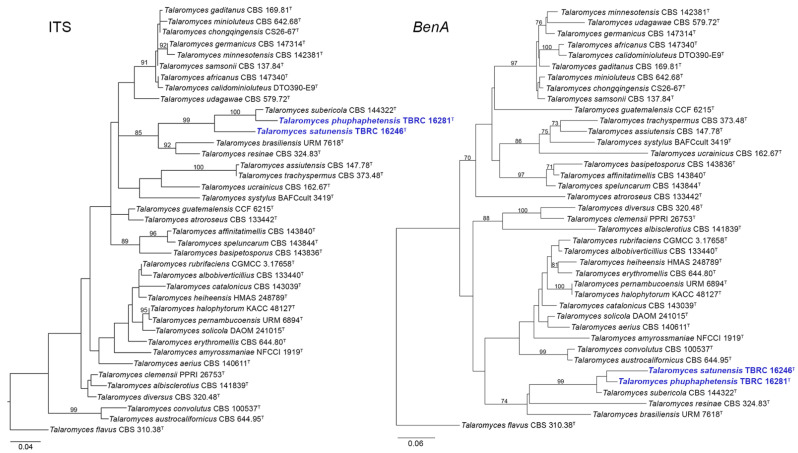
Maximum likelihood phylogeny based on the ITS region (**left**) and the *BenA* gene (**right**) for the closely related species belonging to *Talaromyces* section *Trachyspermi*. *Talaromyces flavus* (CBS 310.38) was chosen as the outgroup. New species are indicated in blue. ^T^ = Ex-type strain. Bootstrap (BS) values ≥ 70% are indicated at the nodes.

**Figure 2 jof-08-00825-f002:**
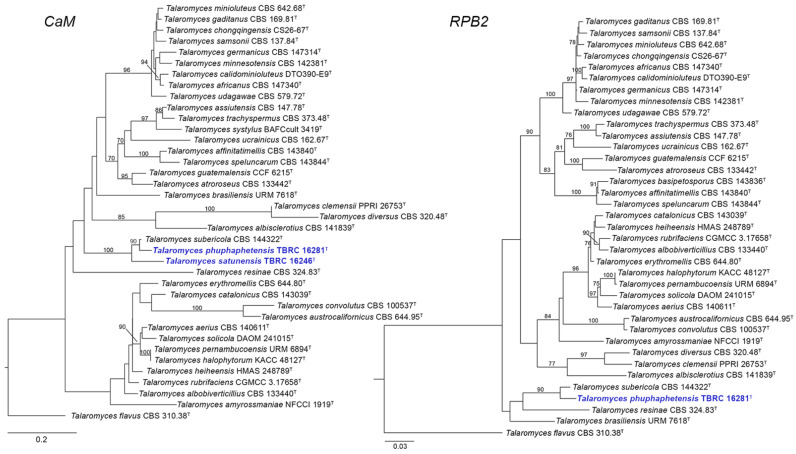
Maximum likelihood phylogeny based on the *CaM* (**left**) and *RPB2* genes (**right**) for the closely related species belonging to *Talaromyces* section *Trachyspermi*. *Talaromyces flavus* (CBS 310.38) was chosen as the outgroup. New species are indicated in blue. ^T^ = Ex-type strain. Bootstrap (BS) values ≥ 70% are indicated at the nodes.

**Figure 3 jof-08-00825-f003:**
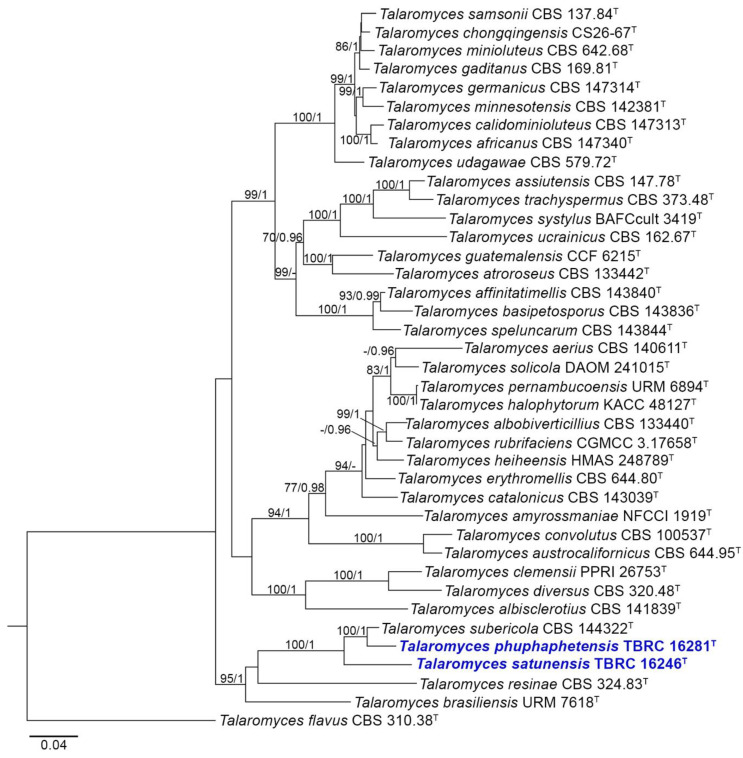
Maximum likelihood phylogeny based on the combination of the ITS region and *BenA*, *CaM,* and *RPB2* genes for the closely related species belonging to *Talaromyces* section *Trachyspermi*. *Talaromyces flavus* (CBS 310.38) was chosen as an outgroup taxon. New species are indicated in blue. ^T^ = Ex-type strain. Bootstrap (BS) values ≥ 70% (**left**) or posterior probability (PP) values ≥ 0.95 (**right**) are indicated at the nodes.

**Figure 4 jof-08-00825-f004:**
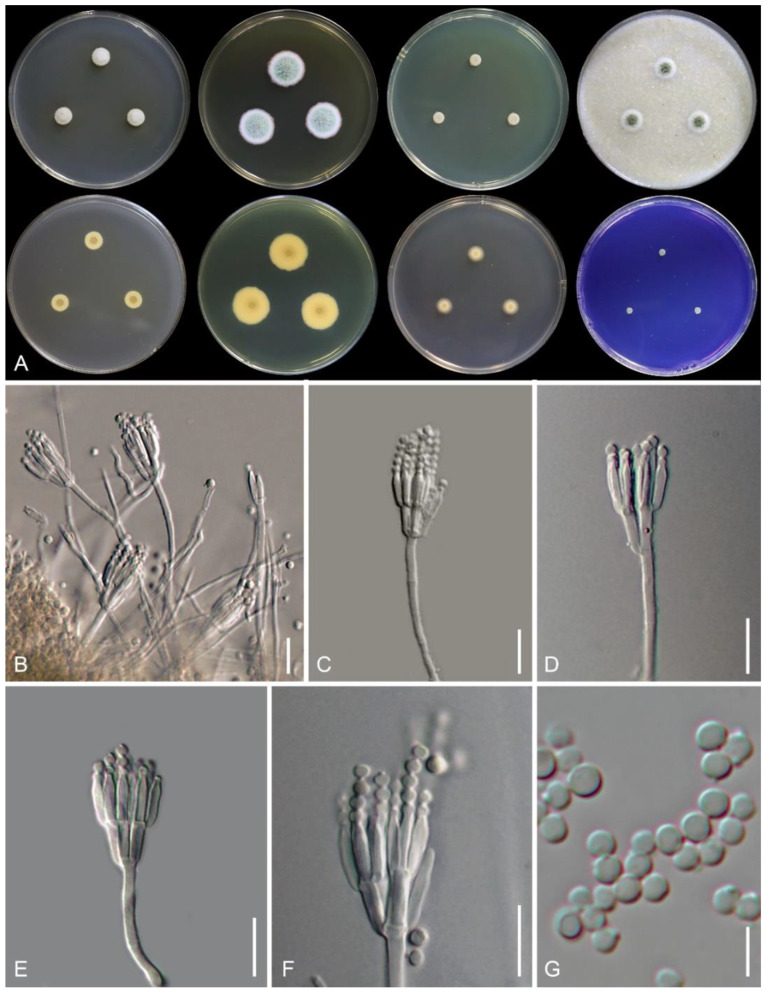
*Talaromyces phuphaphetensis* TBRC 16281. (**A**) Colonies from left to right: (top row) CYA, MEA, YES, and OA, and (bottom row) CYA reverse, MEA reverse, DG18, and CREA. (**B**–**F**) Conidiophores. (**G**) Conidia. Scale bars: (**B**–**F**) = 10 µm, G = 5 µm.

**Figure 5 jof-08-00825-f005:**
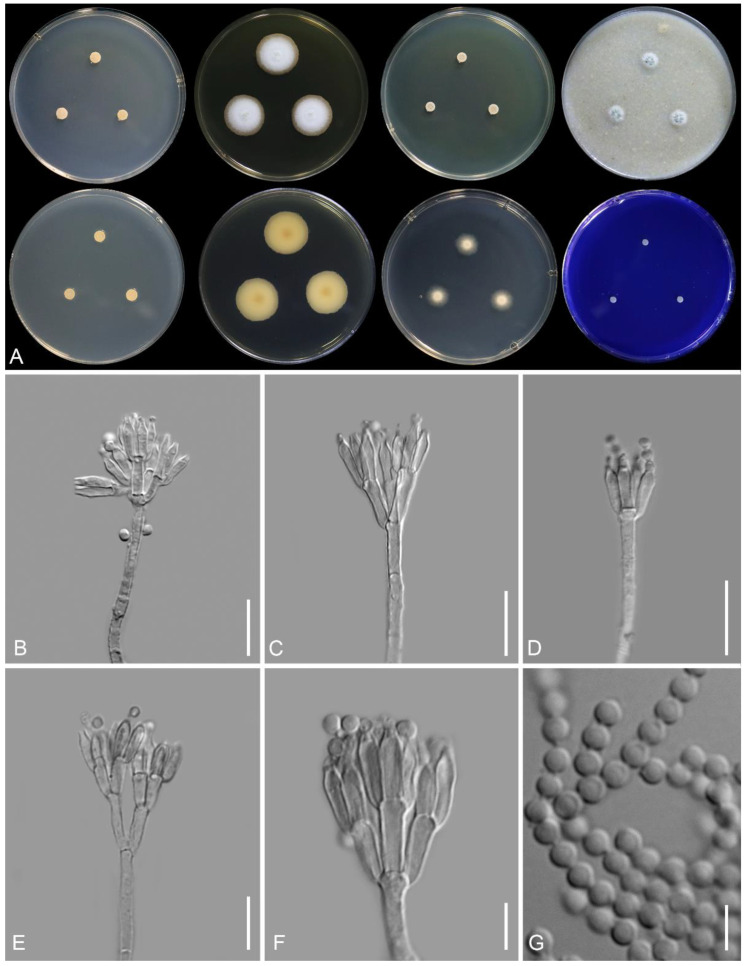
*Talaromyces satunensis* TBRC 16246. (**A**) Colonies from left to right: (top row) CYA, MEA, YES, and OA, and (bottom row) CYA reverse, MEA reverse, DG18, and CREA. (**B**–**F**) Conidiophores. (**G**) Conidia. Scale bars: (**B**–**F**) = 10 µm, G = 5 µm.

**Table 3 jof-08-00825-t003:** Comparisons of the morphological characteristics of *T*. *phuphaphetensis* sp. nov. (TBRC 16281), *T*. *satunensis* sp. nov. (TBRC 16246), and closely related species of *Talaromyces* section *Trachyspermi* based on the phylogeny in this study.

	Microscopic Characters		*T. brasiliensis* [4]	*T. phuphaphetensis* (This Study)	*T. resinae* [21]	*T. satunensis* (This Study)	*T. subericola* [7]
On MEA	Conidiophore	stipes (μm)	20–50 × 2.5–4	15–60 × 2.5–3	N/A	20–290 × 2–3.2	N/A
		branching	biverticillate	mostly biverticillate, monoverticillate		mostly biverticillate, monoverticillate, terverticillate	
		ornamentation	smooth	finely tuberculate		tuberculate	
	Metulae	size (μm)	8–11 × 2.5–3.5	5–9 × 1.5–3		5.5–10 × 2–3.3	
		per verticil	5–6	2–6		2–5	
	Phialides	size (μm)	7–11 (−14) × 2–3	7–9.5 × 2–3		6–9 × 2–3.2	
		per metula	3–4	3–5		2–5	
	Conidia	size (μm)	2–3	2.0–3.5		2.5–3	
		shape	globose	Globose to sub-globose		globose to sub-globose	
		ornamentation	finely roughened	smooth		smooth	
On CZ	Conidiophore	stipes (μm)	N/A	30 − 100 (–120) × 2 − 3	40 − 60 (−80) × 3 − 4	25 − 135 × 2 − 3.5	N/A
		branching		biverticillate, monoverticillate	most biverticillate, monoverticillate symmetric	biverticillate, monoverticillate	
		ornamentation		tuberculate	smooth	tuberculate	
	Metulae	size (μm)		6 − 8 × 2.5 − 3	6–8 (–12) × 2.5–3.5	5−9 × 2−3.5	
		per verticil		2–4	N/A	2–3	
	Phialides	size (μm)		5 − 11 × 2 − 3	6–8 (−12) × 2 − 3	6 − 9 × 2 − 3	
		per metula		3–4	N/A	3–5	
	Conidia	size (μm)		3–4	(3−) 3.5 − 4.5 (−5)	3–4	
		shape		globose to sub-globose	globose to sub-globose	globose to sub-globose	
		ornamentation		smooth	tuberculate	smooth	
On CYA *	Conidiophore	stipes (μm)	N/A	20 − 70 × 2 − 3	N/A	35 − 85 × 2 − 2.5	30 − 45 × 2 − 3
		branching		biverticillate, monoverticillate		biverticillate, monoverticillate	biverticillate
		ornamentation		finely tuberculate		finely tuberculate	smooth
	Metulae	size (μm)		8 − 12 × 2 − 3		6 − 9 × 2 − 2.5	12 − 20 × 2 − 3
		per verticil		2–6		2–4	2–3
	Phialides	size (μm)		8 − 11 × 2 − 3		6.5 − 10.5 × 2 − 2.5	7 − 10 × 2 − 3
		per metula		3–5		3–4	2–4
	Conidia	size (μm)		2–2.5		2–3	3
		shape		globose to sub-globose		globose to sub-globose	ellipsoidal to globose
		ornamentation		smooth		smooth	smooth-walled but verruculose with age

N/A = data not available. * = Microscopic characters were derived after incubation for 2 to 3 weeks.

## Data Availability

All newly generated sequences have been deposited to the GenBank.

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
