# Peer review of "Two Novel Species of Talaromyces Discovered in a Karst Cave in the Satun UNESCO Global Geopark of Southern Thailand"

_jof, 2022, doi:10.3390/jof8080825_

Round 1

Reviewer 1 Report

The study is prepared correctly, I detected no significant irregularities. I have a few questions, as follows:

1. How many soil samples were used in this study? What do you think, would the higher number of soil samples reveal more Talaromyces species? Did you notice only these new Talaromyces species, or also other ones?

2. Please, refer to the methods of the species description. Is the polyphasic approach combining phenotypic and molecular data a standard method or a novel approach in the case of Talaromyces? Please, compare the method proposed by your study with the other studies on Aspergillaceae species. 

3. Is the 29 papers in the bibliography enough? 

4. This study is not on the distribution of Talaromyces species, but I think, the ecology and distribution of the genus Talaromyces are needed to be introduced in the Introduction section. 

5. Please, clarify the potential distribution of these two newly described species on the background of other Talaromyces spp. Are these new species putative common soil fungi, or could be endemic to the cave ecosystems? 

Author Response

Comments and Suggestions for Authors I

The study is prepared correctly, I detected no significant irregularities. I have a few questions, as follows:

  1. How many soil samples were used in this study? What do you think, would the higher number of soil samples reveal more Talaromyces species? Did you notice only these new Talaromyces species, or also other ones?

>> Soil samples were taken from five separate sub-spots and combined into one sample as a representative sample. Also, we mentioned in the text file on page 2 that “Ten or twenty grams of soil were randomly collected at shallow depths (1–5 cm) after removing the surface layer, placed in zip lock bags……”.

According to our findings, we obtained some of Talaromyces species from soil samples, however the most of them are tentatively classified and show that they are not novel candidates (they were only previously identified as common species), and others are being examined.

  1. Please, refer to the methods of the species description. Is the polyphasic approach combining phenotypic and molecular data a standard method or a novel approach in the case of Talaromyces? Please, compare the method proposed by your study with the other studies on Aspergillaceae species.

>> Both our studies on Talaromyces species and fungal genera, as well as some species in the Aspergillaceae family, are most similar in terms of a polyphasic approach combining phenotypic and genetic data. The ITS region, BenA, CaM, and RPB2 genes are frequently analyzed to recognize novel species. Their phylogenetic trees, which are based on morphological data on various-selected synthetic media, multi-loci datasets, and extrolites profiles on CYA or YES in some genera, have been used exclusively to describe not only new taxa but also to show relationships among species in fungal systematics.

  1. Is the 29 papers in the bibliography enough?

>> We are absolutely positive that the 29 papers cited in our paper are sufficient. Dr. Wang, one of the co-authors and a fungal expert, assisted in double-checking and confirming that they are relevant and sufficient.

  1. This study is not on the distribution of Talaromyces species, but I think, the ecology and distribution of the genus Talaromyces are needed to be introduced in the Introduction section.

>> Thank you for your advice. Yes, Talaromyces species are found in a variety of substrates, most notably soil, and in foods such as fruit, nuts, and cereals. All content regarding to the ecology and distribution of the genus Talaromyces is already mentioned on page 1, lines 39-43.

  1. Please, clarify the potential distribution of these two newly described species on the background of other Talaromyces spp. Are these new species putative common soil fungi, or could be endemic to the cave ecosystems?

>> Yes, Talaromyces species can be found in a wide variety of substrates and have been isolated from a variety of materials and hosts, most prominently soil. Our novel taxa are considered common soil yet found in cave habitats in this study. Due to no research on fungal diversity and taxonomy in this cave.  The soil in this cave has never been collected, and anyone interested in conducting research must obtain authorization from the Satun UNESCO Global Geopark. Fortunately, this is described as a novel taxa discovered using polyphasic techniques.

---------------------------------------------------------------------------------

---------------------------------------------------------------------------------

Reviewer 2 Report

Title: Two novel species of Talaromyces discovered in a karst cave in the Satun UNESCO Global Geopark of southern Thailand

 Authors: Salilaporn Nuankaew, Charuwan Chuaseeharonnachai, Sita Preedanon, Sayanh Somrithipol, Supicha Saengkaewsuk, Papichaya Kwantong, Sarinya Phookongchai, Prasert Srikitikulchai, Noppol Kobmoo, Xin- Cun Wang, Zhi-Feng Zhang, Lei Cai, Satinee Suetrong, and Nattawut Boonyuen

 Reference: jof-1853526

 Article type: Research

 Reviewer Comments:

The manuscript jof-1853526, entitled “Two novel species of Talaromyces discovered in a karst cave in the Satun UNESCO Global Geopark of southern Thailand”, identifies cave-dwelling soil fungi discovered from Satun UNESCO Global Geopark (Satun Province, Thailand) using phenotypic approach combined with polyphasic molecular data (ITS, BenA, CaM, and RPB2). The present study, includes the description of two new species - Talaromyces phuphaphetensis spp. nov. and T. satunensis spp. nov. .

 General comments:

- improve overall English grammar and style

- improve the level of scientific writing style (e.g., lines 31-32: to be an unusual environment with high potential for discovering fungi new to science.”

- improve the readability and clarity of the text

- improve consistency

- each idea should be presented in a different paragraph

- limit to a maximum of three the number of references used per sentence (e.g., Line 43: [2–7]), preferring reference works

- use SI unit (e.g., Line 47: 50 rai (80,000 m2))

- number until ten have to be written in full (e.g., Line 70: 2–3 days)

- the Material and Methods should be simplified. For instance, if the methods were used as previously described elsewhere, there is no need to exhaustively described them. The reference is enough.

- Plagiarism check was performed (Abstract: 5%; Introduction: 0%; Material and Methods: 14%; Results: 0%; Discusssion: 0%;

  Specific comments:

Line 26: please consider replacing it with “Based on the four locus datasets”

Lines 27-30: please consider replacing it with “constructed, and two new species  - Talaromyces phuphaphetensis spp. nov. and T. satunensis spp. nov. - phylogenetically related to T. subericola, T. resinae, and T. brasiliensis are described.”

Lines 32 – 33: please consider replacing it with “which appears to be a unique environment with a high potential for discovering fungal species never previously described.”

Lines 40 – 41: please consider replacing it with “Pezizomycotina, and Ascomycota [ref]”

Line 42: please consider replacing it with “air, living or decaying plants”

Lines 51 - 55: please consider replacing it with “In this study, soil samples randomly obtained from the Phu Pha Phet Cave were subjected to phenotype and phylogenetic approaches, and two new cave-dwelling soil microfungi belonging to Talaromyces - T. phuphaphetensis and T. satunensis spp. nov. - were described.”

Line 61: please consider replacing it with “Ten or twenty grams of soil were randomly collected”

Line 63: please consider replacing it with “ice box during collection, and transferred”

Line 71 – 75: please consider replacing it with “After seven days, macroscopic features and growth rates were examined on seven traditional culture media [Czapek yeast autolysate agar (CYA, supplier), Czapek's agar (CZ, supplier), malt extract agar (MEA, Oxoid, Hampshire, UK), yeast extract sucrose agar (YES, supplier), oatmeal agar (OA, Difco), dichloran 18% glycerol Agar (DG18, supplier), and creatine sucrose agar (CREA, supplier)], as previously described [10].”

Lines 79 – 80: please consider replacing it with “Microscopic observations were made on 7-day-old MEA, CZ, and CYA media.”

Lines 81-82: please consider replacing it with “conidiogenous cells, and conidia)

Line 84: please consider replacing it with “Methuen Handbook of Color created color codes”

Line 97: please consider replacing it with “(CaM), and RNA polymerase II”

Lines 103 – 104: please consider replacing it with “section Trachyspermi used in phylogenetic analyses and their accession numbers”

Line 107: please consider replacing it with “analyses, including 1000 bootstrap replicates, were”

Lines 110 – 111: please consider replacing it with “consensus tree was constructed using FigTree v1.4.4”

Line 135: please consider replacing it with “ITS, BenA, CaM, and RPB2”

Lines 139 - 140: please consider replacing it with “were clustered with T. brasiliensis URM 7618, T. resinae CBS 324.83, and T. subericola CBS 144322”

Line 141: please consider replacing it with “T. subericola formed a monophyletic group, phylogenetically related to T. brasiliensis

Lines 146 – 148: please clarify the sentence. Was the sequence from RPB2 from T. satunensis not sequenced. Why?

Line 149: please consider replacing it with “CaM, and RPB2, the phylogenetic”

Line 150: please consider replacing it with “obtained for each gene individually (Figure 3).”

Line 150: please consider replacing it with “The two new species, T. phuphaphetensis and T. satunensis, formed”

Lines 283-284: please consider replacing it with “Phylogenetic analysis based on a single gene (i.e., ITS, BenA, CaM, and RPB2) and the polygenic approach showed”

Line 285: please consider replacing it with “T. resinae, and T. subericola.”

Line 286: please consider replacing it with “T. satunensis, and T. subericola (Figure 3).”

Line 292: please consider replacing it with “are congruent with Rajeshkumar et al. [5]”

Line 295: please consider replacing it with “locus for identifying Talaromyces species”

Lines 296-298: please consider replacing it with “Unfortunately, this study did not obtain sequence data from T. satunensis for the RPB2 gene, although we attempted with different PCR profiles.”

Line 300: please consider replacing it with “as two distinct species”

Line 303: please consider replacing it with “tuberculate-walled stipes, and smooth-walled conidia”

Line 305: please consider replacing it with “These data agree with the section's description [25–26].”

Line 306: please consider replacing it with “yellow pigment on CYA, CZ, and DG18.”

Lines 307 - 309: please consider replacing it with “Many species in section Trachyspermi (such as T. albobiverticillius, T. atroroseus, and T. minioluteus) can produce large amounts of red pigment.”

Line 310: please consider replacing it with “produces pigments without known mycotoxins”

Lines 313 - 314: please consider replacing it with “pigment production, and another testing for future research.”

Line 319: please consider replacing it with “differences from the species described in section Trachyspermi.”

Lines 326 - 327: please consider replacing it with “Interestingly, it is also possible that more new species will be discovered in this peculiar environment in Thailand's Satun UNESCO Global Geopark.”

Author Response

Comments and Suggestions for Authors II

Reviewer Comments:

The manuscript jof-1853526, entitled “Two novel species of Talaromyces discovered in a karst cave in the Satun UNESCO Global Geopark of southern Thailand”, identifies cave-dwelling soil fungi discovered from Satun UNESCO Global Geopark (Satun Province, Thailand) using phenotypic approach combined with polyphasic molecular data (ITS, BenA, CaM, and RPB2). The present study, includes the description of two new species - Talaromyces phuphaphetensis spp. nov. and T. satunensis spp. nov. .

General comments:

- improve overall English grammar and style

>> We have sent it to AJE, thus, please see the updated version supporting with the verification code 0460-FB32-2D43-CCBB-773P at AJE (https://www.aje.com).

- improve the level of scientific writing style (e.g., lines 31-32: to be an unusual environment with high potential for discovering fungi new to science.”

>> We revised it as yellow texts

- improve the readability and clarity of the text

We have improved it as an updated version, based on suggested content and English improvement by AJE.

- improve consistency

We have improved it as an updated version, based on suggested content and English improvement by AJE.

- each idea should be presented in a different paragraph

>> Thank you for the suggestion, we have improved them by presenting each idea per paragraph.

- limit to a maximum of three the number of references used per sentence (e.g., Line 43: [2–7]), preferring reference works

>> All citations support all fungal-based content in various of habitats, including soil, vegetation, air, live or decaying plants, indoor environments, and food products. As a result, we still remain all references as no. 2 - no. 7.

- use SI unit (e.g., Line 47: 50 rai (80,000 m2))

>> We revised it as yellow texts as SI unit

- number until ten have to be written in full (e.g., Line 70: 2–3 days)

>> We revised it as yellow texts

- the Material and Methods should be simplified. For instance, if the methods were used as previously described elsewhere, there is no need to exhaustively described them. The reference is enough.

>> Thank you for your advice; however, even if it has already been mentioned, we would supply all relevant information as texts. One of our hidden aims for support texts is that no author has to spend time checking them and that it is not necessary to spend time finding them out in detail.

- Plagiarism check was performed (Abstract: 5%; Introduction: 0%; Material and Methods: 14%; Results: 0%; Discusssion: 0%;

> Thank you for double checking of this data

Specific comments:

Line 26: please consider replacing it with “Based on the four locus datasets”

>> We revised it as yellow texts

Lines 27-30: please consider replacing it with “constructed, and two new species  - Talaromyces phuphaphetensis spp. nov. and T. satunensis spp. nov. - phylogenetically related to T. subericola, T. resinae, and T. brasiliensis are described.”

>> We revised it as yellow texts

Lines 32 – 33: please consider replacing it with “which appears to be a unique environment with a high potential for discovering fungal species never previously described.”

>> We revised it as yellow texts

Lines 40 – 41: please consider replacing it with “Pezizomycotina, and Ascomycota [ref]”

>> We revised it as yellow texts

Line 42: please consider replacing it with “air, living or decaying plants”

>> We revised it as yellow texts

Lines 51 - 55: please consider replacing it with “In this study, soil samples randomly obtained from the Phu Pha Phet Cave were subjected to phenotype and phylogenetic approaches, and two new cave-dwelling soil microfungi belonging to Talaromyces - T. phuphaphetensis and T. satunensis spp. nov. - were described.”

>> We revised it as yellow texts

Line 61: please consider replacing it with “Ten or twenty grams of soil were randomly collected”

>> we revised it as yellow texts

Line 63: please consider replacing it with “ice box during collection, and transferred”

>> We revised it as yellow texts

Line 71 – 75: please consider replacing it with “After seven days, macroscopic features and growth rates were examined on seven traditional culture media [Czapek yeast autolysate agar (CYA, supplier), Czapek's agar (CZ, supplier), malt extract agar (MEA, Oxoid, Hampshire, UK), yeast extract sucrose agar (YES, supplier), oatmeal agar (OA, Difco), dichloran 18% glycerol Agar (DG18, supplier), and creatine sucrose agar (CREA, supplier)], as previously described [10].”

>> We self-prepared seven media, referencing Visagie et al (2014). We made minor modifications to OA by modifying oat meal agar (difco) replaced oat flakes and adding trace elements (zinc-sulphate and copper-sulphate).

Lines 79 – 80: please consider replacing it with “Microscopic observations were made on 7-day-old MEA, CZ, and CYA media.”

>> We revised it as yellow texts

Lines 81-82: please consider replacing it with “conidiogenous cells, and conidia)

>> We revised it as yellow texts

Line 84: please consider replacing it with “Methuen Handbook of Color created color codes”

>> We revised it as yellow texts

Line 97: please consider replacing it with “(CaM), and RNA polymerase II”

>> we revised it as yellow texts

Lines 103 – 104: please consider replacing it with “section Trachyspermi used in phylogenetic analyses and their accession numbers”

>> We revised it as yellow texts

Line 107: please consider replacing it with “analyses, including 1000 bootstrap replicates, were”

>> We revised it as yellow texts

Lines 110 – 111: please consider replacing it with “consensus tree was constructed using FigTree v1.4.4”

>> We revised it as yellow texts

Line 135: please consider replacing it with “ITS, BenA, CaM, and RPB2”

>> We revised it as yellow texts

Lines 139 - 140: please consider replacing it with “were clustered with T. brasiliensis URM 7618, T. resinae CBS 324.83, and T. subericola CBS 144322”

>> We revised it as yellow texts

Line 141: please consider replacing it with “T. subericola formed a monophyletic group, phylogenetically related to T. brasiliensis”

>> We revised it as yellow texts

Lines 146 – 148: please clarify the sentence. Was the sequence from RPB2 from T. satunensis not sequenced. Why?

>> The lack of RPB2 data has been explained in the discussion section. Despite numerous attempts, PCR and sequencing unsuccessful.

Line 149: please consider replacing it with “CaM, and RPB2, the phylogenetic”

>> We revised it as yellow texts

Line 150: please consider replacing it with “obtained for each gene individually (Figure 3).”

>> We revised it as yellow texts

Line 150: please consider replacing it with “The two new species, T. phuphaphetensis and T. satunensis, formed”

>> We revised it as yellow texts

Lines 283-284: please consider replacing it with “Phylogenetic analysis based on a single gene (i.e., ITS, BenA, CaM, and RPB2) and the polygenic approach showed”

>> we revised it as yellow texts

Line 285: please consider replacing it with “T. resinae, and T. subericola.”

>> We revised it as yellow texts

Line 286: please consider replacing it with “T. satunensis, and T. subericola (Figure 3).”

>> We revised it as yellow texts

Line 292: please consider replacing it with “are congruent with Rajeshkumar et al. [5]”

>> We revised it as yellow texts

Line 295: please consider replacing it with “locus for identifying Talaromyces species”

>> We revised it as yellow texts

Lines 296-298: please consider replacing it with “Unfortunately, this study did not obtain sequence data from T. satunensis for the RPB2 gene, although we attempted with different PCR profiles.”

>> We revised it as yellow texts

Line 300: please consider replacing it with “as two distinct species”

>> We revised it as yellow texts

Line 303: please consider replacing it with “tuberculate-walled stipes, and smooth-walled conidia”

>> We revised it as yellow texts

Line 305: please consider replacing it with “These data agree with the section's description [25–26].”

>> We revised it as yellow texts

Line 306: please consider replacing it with “yellow pigment on CYA, CZ, and DG18.”

>> We revised it as yellow texts

Lines 307 - 309: please consider replacing it with “Many species in section Trachyspermi (such as T. albobiverticillius, T. atroroseus, and T. minioluteus) can produce large amounts of red pigment.”

>> We revised it as yellow texts

Line 310: please consider replacing it with “produces pigments without known mycotoxins”

>> We revised it as yellow texts

Lines 313 - 314: please consider replacing it with “pigment production, and another testing for future research.”

>> We revised it as yellow texts

Line 319: please consider replacing it with “differences from the species described in section Trachyspermi.”

>> We revised it as yellow texts

Lines 326 - 327: please consider replacing it with “Interestingly, it is also possible that more new species will be discovered in this peculiar environment in Thailand's Satun UNESCO Global Geopark.”

>> We revised it as yellow texts
